# Path Analysis of the Main Control Factors of Transpiration in Greenhouse, Drip-Irrigated Grapes in Cold Areas of Northeast China

**Dongjie Pei, Xinguang Wei, Yikui Bai \*, Cong Wang** 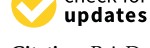 **, Ying Liu and Senyan Jiang**

College of Water Conservancy, Shenyang Agricultural University, Shenyang 110866, China
\* Correspondence: baiyikui@syau.edu.cn; Tel.: +86-133-8683-4288

**Abstract:** To investigate the characteristics of grape transpiration water consumption and its environmental coupling mechanism in a greenhouse growing environment in cold areas of Northeast China, the dynamic monitoring of greenhouse grape sap flow and microenvironmental factors in a greenhouse was carried out for two years. Correlation analysis and path analysis were used to study the characteristics of grape transpiration environmental factors at different temporal scales (instantaneous, daily, and growth period) and the influence mechanisms on greenhouse grape transpiration. The results of correlation analysis by growth period showed that, on the instantaneous scale, the correlation between each meteorological factor and grape transpiration reached a significant level (coefficient of determination $R^2$ ranged from 0.25 to 0.84). On the daily scale, the correlation of solar radiation ($R_s$) was the best except for the new growth period ($R^2$ ranged from 0.49 to 0.89). The results of the split-fertility path analysis showed that the total effects of $R_s$ on instantaneous transpiration were the largest at all stages of fertility, with decision coefficients ($R$) ranging from 0.69 to 0.90. On the daily scale, the total and direct effects of $R_s$ on daily transpiration were the largest ($R$ ranged from 0.70 to 0.94), except for the new growth period. The results of the whole growth period path analysis showed that $R_s$ had the greatest effect on instantaneous transpiration, with $R$ of 0.86. On the daily scale, $R_s$ was also the most influential factor in grape transpiration, with $R$ of 0.81. On the growth period scale, only air temperature ($T_a$) and leaf area index (LAI) were significantly correlated with transpiration ($p < 0.05$), and $R_s$ had the largest total effect on transpiration with $R$ of 0.68. To sum up, on each time scale, $R_s$ was always the most important factor influencing grape transpiration. However, as the time scale increased, the effects of LAI and soil water content (SW) on transpiration increased while the effects of $R_s$, $T_a$, RH, and VPD on transpiration gradually decreased.

**Keywords:** transpiration; correlation analysis; path analysis; scale effect



## 1. Introduction

Transpiration is not only a major pathway for water consumption by crops but also an important link in the soil–plant–atmosphere continuum [1–3]. It also plays a key role in irrigation water use [4]. By reasonably controlling the frequency and amount of irrigation, the yield and quality of crops can be significantly improved [5,6]. Previous studies on water consumption by transpiration in greenhouses have focused on water consumption characteristics, irrigation systems, and mechanisms and patterns of efficient water and fertilizer use [7–9]. Among them, the research on water consumption characteristics, main environmental control factors, and their regulatory mechanisms at different spatial and temporal scales of crops is the basis for precise crop water management [10].

Nalevanková et al. [11] used principal component analysis to study the transpiration and environmental shadow of European beech, and the results showed that daily transpiration was closely related to solar radiation, air temperature, and relative humidity, and seasonal transpiration was closely related to soil water deficit. Xu et al. [12] used stepwise linear regression analysis to investigate the effects of environmental variables on desert shrub transpiration, and the results showed that solar radiation was the main driver of evapotranspiration. Zhang

et al. [13] studied the spatial scale effect of evapotranspiration in summer maize using path analysis and showed that net radiation was always the main influence on evapotranspiration at leaf, monocot, and farm scales. Han et al. [14] used regression analysis to study evapotranspiration in savannas, and the results showed that evapotranspiration was controlled by net radiation on an instantaneous scale while evapotranspiration was controlled by leaf area index and soil moisture content on a daily scale. Sun et al. [15] used correlation analysis to analyze evapotranspiration in different ecosystems and showed that leaf area index and air temperature were the main controlling factors in determining evapotranspiration at the growing season and annual scales. Pour et al. [16] conducted a sensitivity analysis on rice evapotranspiration in Peninsular Malaysia. The results showed that minimum air temperature was the main meteorological factor affecting daily rice evapotranspiration. Liu et al. [17] used sensitivity analysis to quantify the relationship between meteorological factors and reference evapotranspiration in Beijing, and the results showed that at the daily scale, net radiation and relative humidity dominated. Qiao et al. [18] used correlation analysis to analyze the transpiration characteristics of date palm trees, and the results showed that leaf area index was the main influencing factor of transpiration at the growth period scale. Li et al. [19] studied the time-scale effect of transpiration in white hairy poplar by path analysis, and the results showed that the main influencing factors of transpiration at the instantaneous scale were solar radiation and vapor pressure deficit, while the main influencing factors of transpiration at the daily scale were solar radiation and air temperature. Cai et al. [20] used path analysis to study the spatio-temporal scale effect of winter wheat evapotranspiration. The results showed that net radiation was the main controlling factor of winter wheat evapotranspiration on a daily scale; leaf area index and underlying vegetation height were the main factors controlling evapotranspiration at the plot and field scales, respectively. Su et al. [21] used correlation analysis to show that temperature was the main influencing factor of evapotranspiration at the temporal and daily scales. Anandacoomaraswamy et al. [22] found that the instantaneous transpiration of tea plants determines soil moisture content and solar radiation. Mellander et al. [23] found that soil temperature was the main control factor affecting the transpiration of forest ecosystems in cold regions. Alvarez et al. [24] studied calla lily and found that soil water stress was the main controlling factor affecting daily transpiration. Albertoa et al. [25], Irmark et al. [26], and Granier et al. [27] analyzed the main influencing factors of crop transpiration, and the results showed that at different scales, the main controlling factor of transpiration was solar radiation followed by saturated water pressure difference.

Water management is an important aspect of growing greenhouse crops. Under greenhouse growing conditions, in order to reduce air humidity, diseases, and insect pests, irrigation is mainly implemented in the form of under-membrane drip irrigation [28], and the evaporative water consumption is almost zero, so the water consumption mode of the greenhouse crop is mainly transpiration [29]. Li et al. [30] used correlation and path analysis to investigate the daily transpiration of greenhouse melon and its influencing factors, and the results showed that the magnitude of influence of each factor on the transpiration of greenhouse melon was leaf area index > average daily temperature > average daily relative air humidity > solar radiation intensity. Li et al. [7] used regression analysis to study the relationship between the characteristics of sap flow and the main environmental influencing factors for greenhouse grapes in cold regions of northeast China, and the results showed that the main influencing factors for instantaneous and daily evapotranspiration were photosynthetically effective radiation and vapor pressure deficit, and the main influencing factor for month-by-month and whole growth period evapotranspiration was photosynthetically effective radiation. Gong et al. [10] studied the spatial scale effect of evapotranspiration in tomatoes in greenhouse using path analysis, and the results showed that solar radiation was the main control factor of evapotranspiration at leaf, single plant, and population scales. Previous studies on the spatial and temporal scale effects of crop water consumption characteristics have mainly focused on field crops. However, systematic studies on water consumption and the main influencing factors at different temporal scales (instantaneous, daily, monthly, decadal, reproductive stage, full growth period, etc.) of

crops in greenhouse, especially those on crop water consumption and environmental regulation and its scale effects in cold and cold regions under heliophile growing conditions, are not available.

In this paper, through systematic analysis of transpiration and its related influencing factors of greenhouse grapes on different time scales, the main influencing factors and influence degrees of grape transpiration on each time scale are explored and the time scale effects are clarified. The research results can provide a reference for the formulation of irrigation systems and the regulation of greenhouse environments.

## 2. Materials and Methods

### 2.1. Study Site

The experiment was carried out from 1 April 2020 to 31 October 2021 in the solar greenhouse No. 44 of Shenyang Agricultural University Research and Experimental Base in China (123°57 ′E, 42°82 ′N) with an east–west direction, span of 8 m, ridge height of 4 m, and length of 60 m. The greenhouse was a Chinese Liaoshen III (ZGLSSG-III) solar greenhouse. The indoor temperature could be adjusted through the vents at the top and bottom. The maximum opening of the top vent was 50 cm. The opening could be automatically adjusted. There was a rainproof quilt driven by a motor on the top. During the test, when the ambient temperature was lower than 16 °C, the quilt was put down at night for insulation.

A total of 111 five-year-old *Vitis vinifera* L. *cv. Muscat Ham-burg* grapes planted in 2016 were used as test materials, with plant spacing of 0.5 m and row spacing of 4.7 m. Israel Netafim drip irrigation pipe was selected as the drip irrigation pipe with a diameter of 16 mm. The dripper adopted a pressure compensation dripper with a spacing of 0.3 m. The designed flow rate of the dripper was 8 L h$^{-1}$. The upper and lower limits of irrigation were 90%$\theta_f$ and 60%$\theta_f$ ($\theta_f$ is the field water holding rate, cm$^3$ cm$^{-3}$), and the fertilizer application rate N, P$_2$O$_5$, K$_2$O was 260, 119, 485 kg hm$^{-2}$, respectively. The test soil was a medium loam with a planned wetting layer depth of 60 cm and a capacity of 1.44 g cm$^{-3}$ from 0 to 60 cm. Other field agronomic management measures were consistent (e.g., pruning of branching stems and pest control) and were based on local experience with grape production. The overview of the test area is shown in Figure 1. Two years of growth periods and transpiration are shown in Table 1.

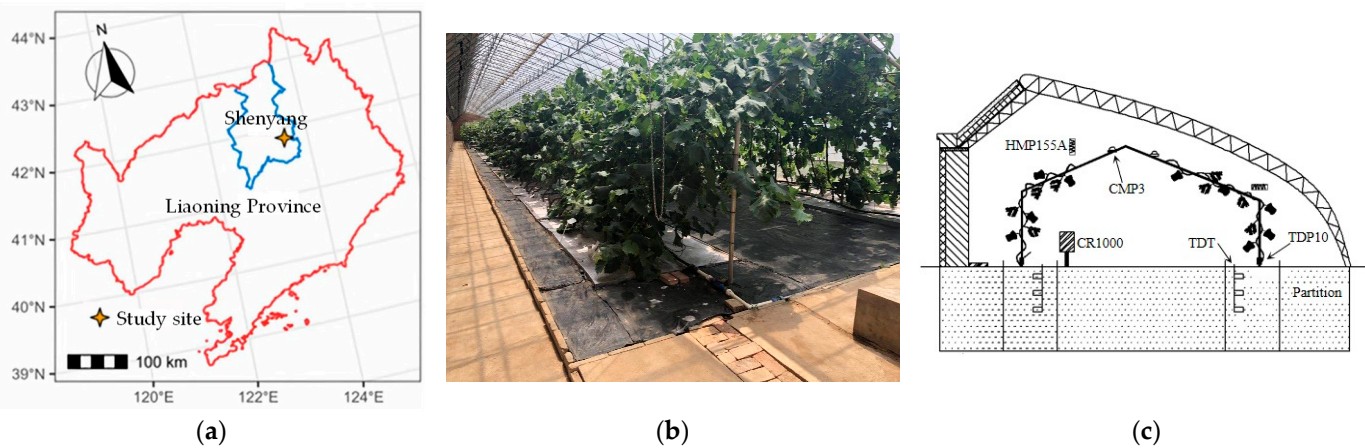

(a)  (b)  (c)

**Figure 1.** Experimental planting area of drip irrigation under plastic film and sensor layout. (**a**) Study site; (**b**) experimental planting area of drip irrigation under plastic film; (**c**) greenhouse structure and sensor layout.

**Table 1.** Growth period division and transpiration in 2020 and 2021.

| Growth Period | 2020 | | 2021 | |
|---|---|---|---|---|
| | Date | Transpiration (mm) | Date | Transpiration (mm) |
| New growth period | 4.11~5.2 | 35.37 | 4.6~4.29 | 22.75 |
| Flowering and fruit setting period | 5.3~5.28 | 38.82 | 4.30~5.25 | 34.95 |
| Fruit expansion period | 5.29~7.12 | 147.27 | 5.26~7.15 | 138.39 |
| Fruit maturation period | 7.13~8.25 | 110.59 | 7.16~8.31 | 109.95 |

*2.2. Data Collection*

Meteorological indicators: a small environmental element monitoring system including a data collector (CR1000, Campbell Scientific, USA), temperature and humidity sensor (HMP155A, Vaisala, Finland), and solar radiation sensor (CMP3, Kipp & Zonen, Netherlands) with a data collection frequency of 10 min. The main monitoring elements were solar radiation ($R_s$, W m$^{-2}$), air temperature ($T_a$, °C), and relative humidity (RH, %). The vapor pressure deficit (VPD, kPa) was calculated according to the method provided by Allen [31].

Soil water content: Soil water content (SW, %) was monitored in real time using a soil moisture sensor (time domain moisture, TDT) (Campbell Scientific, USA) with a collection frequency of 10 min. In this study, soil moisture sensors were deployed at the depths of 15, 30, and 45 cm respectively, and the average value of 3 sensors was taken as the soil moisture content value.

Leaf area index: The total leaf area of the corresponding grape plants was estimated using a random sampling method, and the leaf area index (LAI) of the plant was obtained by dividing the leaf area projection by the canopy projection area. The measurement frequency was 7 to 10 d.

Sap flow rate and transpiration: A set of thermal dissipation probes (TDP10) were installed 20 cm from the ground in each of the selected vines, and the collection frequency was 10 min. The transpiration rate was calculated as follows:

$$F_d = 118.99 \times 10^{-6} [(\Delta V_{max} - \Delta V)/\Delta V]^{1.231} \tag{1}$$

$$F_s = F_d \times A_s \times 3600 \tag{2}$$

$$Tr_d = \frac{1}{6} \sum_{i=1}^{144} \frac{F_{si}}{1000A} \tag{3}$$

$$Tr_p = \sum_{i=1}^{n} Tr_{di} \tag{4}$$

where $F_d$ is the sap flow rate, m s$^{-1}$; $\Delta V_{max}$ is the thermodynamic potential of the thermocouple when the sap flow rate is zero, V; $\Delta V$ is the thermodynamic potential of the thermocouple in the presence of sap flow, V; $F_s$ is the instantaneous transpiration, g h$^{-1}$; $A_s$ is the cross-sectional area of the water-conducting sapwood section, m$^2$; $Tr_d$ is the daily transpiration, mm d$^{-1}$; A is the grape shade area, m$^2$; $Tr_p$ is the transpiration during the growth period, mm; and n is the number of days in the growth period, d.

*2.3. Path Analysis Method*

The path analysis method was proposed by Sewall Wright [32] in 1921 and has been refined by scholars to form a multivariate statistical method. It investigates the relative importance of the dependent variable by decomposing the independent and dependent variables into each other. This method elucidates the direct and indirect effects of all factors related to the dependent variable to determine the multivariate linear equation. For an

interrelated system, if there is a linear relationship between the n independent variables $X_i$ and the dependent variable $Y$, the regression equation is

$$Y = b_0 + b_1 X_1 + b_2 X_2 + \cdots + b_n X_n \tag{5}$$

According to the simple correlation coefficient $r_{ij}$ $(i, j \leq n)$ between the respective variables and the simple correlation coefficient $r_{iy}$ between the respective variables and the dependent variable, the matrix equation is obtained by transforming Equation (5),

$$\begin{bmatrix} 1 & r_{12} & \cdots & r_{1n} \\ r_{21} & 1 & \cdots & r_{2n} \\ \vdots & \vdots & \ddots & \cdots \\ r_{n1} & r_{n2} & \cdots & 1 \end{bmatrix} \begin{bmatrix} P_{y1} \\ P_{y2} \\ \vdots \\ P_{yn} \end{bmatrix} = \begin{bmatrix} r_{1y} \\ r_{2y} \\ \vdots \\ r_{ny} \end{bmatrix} \tag{6}$$

where $P_{yi}$ is the direct path coefficient of the independent variable $X_i$ on the dependent variable $Y$, i.e., the direct effect of $X_i$ on $Y$. $r_{ij}P_{yi}$ is the indirect path coefficient of the independent variable $X_i$ on $Y$ through $X_j$, i.e., the indirect path coefficient of $X_i$ on $Y$ through pairs of $X_j$.

## 3. Results

### 3.1. Dynamics of Transpiration and Environmental Factors in Grapes throughout the Growth Period

As can be seen in Figure 2, daily transpiration ($Tr_d$), solar radiation ($R_s$), and temperature ($T_a$) of grapes showed an overall unimodal trend throughout the whole growth period. The maximum $R_s$ appeared in June, and the maximum daily $R_s$ in 2020 and 2021 were 203 W m$^{-2}$ (14 June) and 218 W m$^{-2}$ (24 June), respectively. In 2020 and 2021, the transpiration reached 332 mm and 306 mm, respectively. The maximum $Tr_d$ occurred in July, and the maximum $Tr_d$ in 2020 and 2021 were 3.02 mm (9 July) and 2.89 mm (7 July), respectively. The maximum $T_a$ occurred in late July with a maximum daily average $T_a$ value of 30.6 °C and 31.1 °C in 2020 and 2021, respectively. LAI showed a continuous increase throughout the whole growth period with a faster increase in April and May and a slower increase in the later part of the whole growth period. RH and VPD fluctuated greatly during the whole growth period in 2020 and 2021. The average daily RH varied between 34.0% and 84.6%, 34.9%, and 88.1%, respectively. The average daily VPD varied between 0.23 kPa and 2.08 kPa, 0.19 kPa~1.98 kPa in 2020 and 2021, respectively. SW varied between 20.5% and 32.8% in two years.

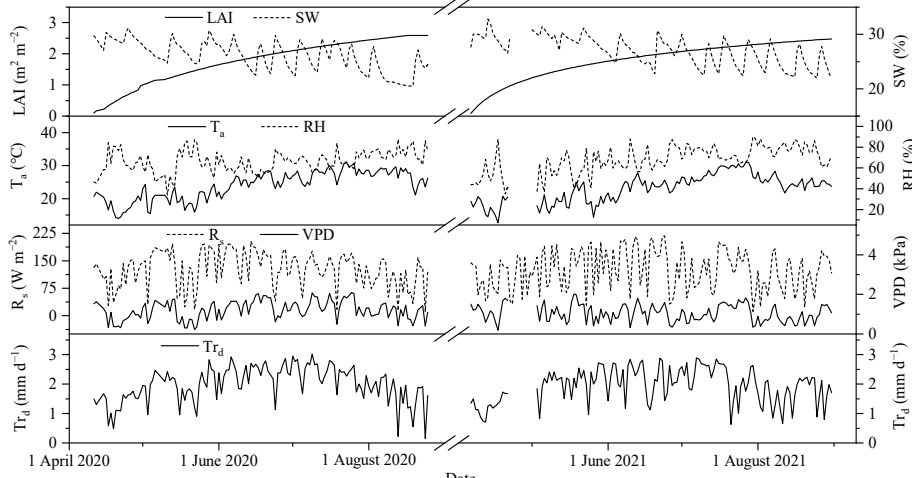

**Figure 2.** Dynamic changes of daily transpiration ($Tr_d$), solar radiation ($R_s$), air temperature ($T_a$), relative humidity (RH), vapor pressure deficit (VPD), soil water content (SW), and leaf area index (LAI).

### 3.2. Correlation Analysis between Grape Transpiration and Environmental Factors

To investigate the relationship between grape growth status, greenhouse environmental factors, and grape transpiration, the correlations between instantaneous transpiration ($F_s$), daily transpiration ($Tr_d$), and each factor were analyzed for different growth periods of greenhouse grapes. On the instantaneous scale (Figure 3), $R_s$, $T_a$, and VPD showed significant positive correlations ($p < 0.01$) with $F_s$. While RH showed a significant negative correlation ($p < 0.01$) with $F_s$ at each growth period. The effect of meteorological factors on instantaneous transpiration differed among the different growth periods. The two-year data showed that the fruit maturation period had the greatest correlation with each meteorological factor ($R^2$ is between 0.67 and 0.84), with the highest correlation between $F_s$ and meteorological factors ($R^2$ between 0.67 and 0.84), followed by the fruit expansion period. The new growth period and the flowering and fruit setting period were less correlated with $F_s$. Among the environmental factors, $R_s$ and $F_s$ had the highest correlation, and their $R^2$ fluctuated between 0.78 and 0.84, 0.49 and 0.84 in 2020 and 2021, respectively. The maximum $R^2$ reached 0.84 at the fruit maturation period in both years. $T_a$, VPD, and $F_s$ also showed good positive correlations. RH and $F_s$ had the lowest $R^2$, especially in the flowering and fruit setting period, with only 0.38 and 0.21 in 2020 and 2021. In general, on the instantaneous scale, the effects of environmental factors and instantaneous transpiration were the most significant during the fruit maturation period, and the effect of $R_s$ was the most significant. The correlation of each factor on the instantaneous transpiration scale was $R_s$ > VPD > $T_a$ > RH.

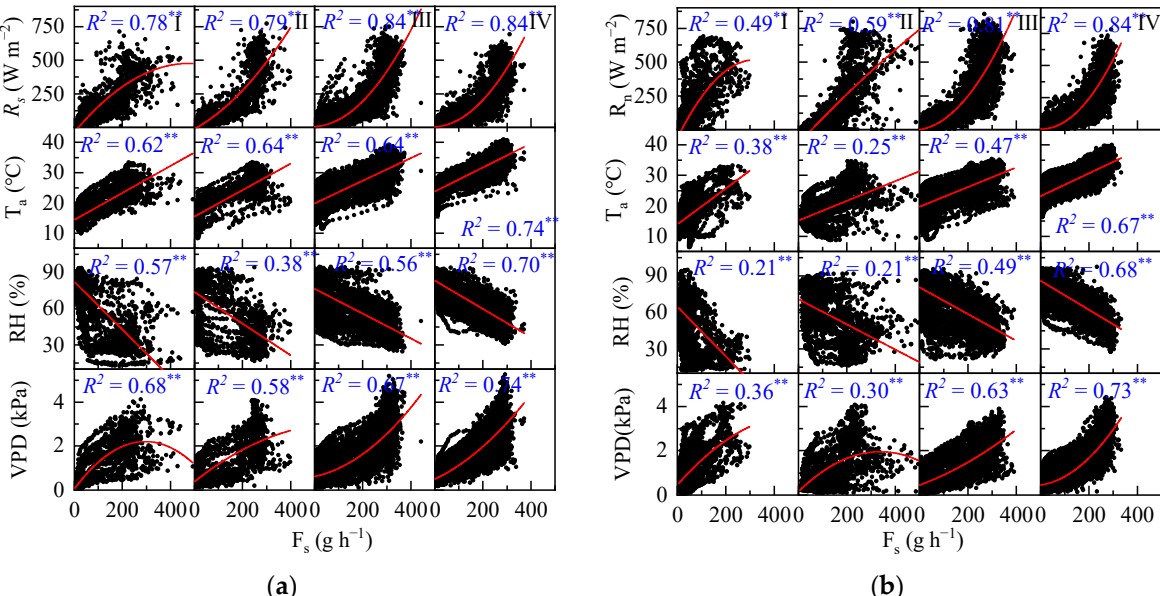

**Figure 3.** Correlation analysis between instantaneous transpiration ($F_s$) and environmental factors. (**a**) 2020; (**b**) 2021. * Indicates significant correlation between independent variable and dependent variable ($p < 0.05$); ** Indicates highly significant correlation between independent variables and dependent variables ($p < 0.01$). I, II, III, and IV respectively represent the new growth period, flowering and fruit setting period, fruit expansion period, and fruit maturation period.

On the daily scale (Figure 4), except for $T_a$ reaching significant correlations ($p < 0.05$) at the flowering and fruit setting period and fruit expansion period in 2021, the effects of $Tr_d$ reached highly significant levels ($p < 0.01$) for all growth periods of $R_s$, RH, VPD, and LAI. However, the SW was not significant with $Tr_d$. Among the meteorological factors, $R_s$ and $Tr_d$ had the best correlation, and $R^2$ for the four growth periods in 2020 and 2021 is between 0.49 and 0.89, with the overall performance of fruit maturation period > fruit expansion > flowering and fruit setting period > new growth period. The correlation between $T_a$ and $Tr_d$ is good; $R^2$ in each growth period over the two years varies from 0.06

to 0.48. The correlation between $T_a$ and $Tr_d$ in the flowering and fruit setting period and fruit expansion period is lower than that in the new growth period and fruit maturation period. The correlation between RH and $Tr_d$ varied from 0.43 to 0.80. The correlation between VPD and $Tr_d$ varied from 0.51 to 0.80. $R^2$ of LAI and $Tr_d$ ranged from 0.21 to 0.44. The four growth periods were flowering and fruit setting period > new growth period > fruit expansion period > fruit maturation period. As can be seen from Figure 4, SW was not correlated with $Tr_d$, which may be due to the fact that water was not a limiting factor for grape transpiration due to the relatively adequate water supply in the vineyard during the test cycle. Overall, the correlations between meteorological factors and daily transpiration were the highest during the fruit maturation period, except for SW and LAI, with the correlations of each factor showing $R_s$ > VPD > RH > $T_a$.

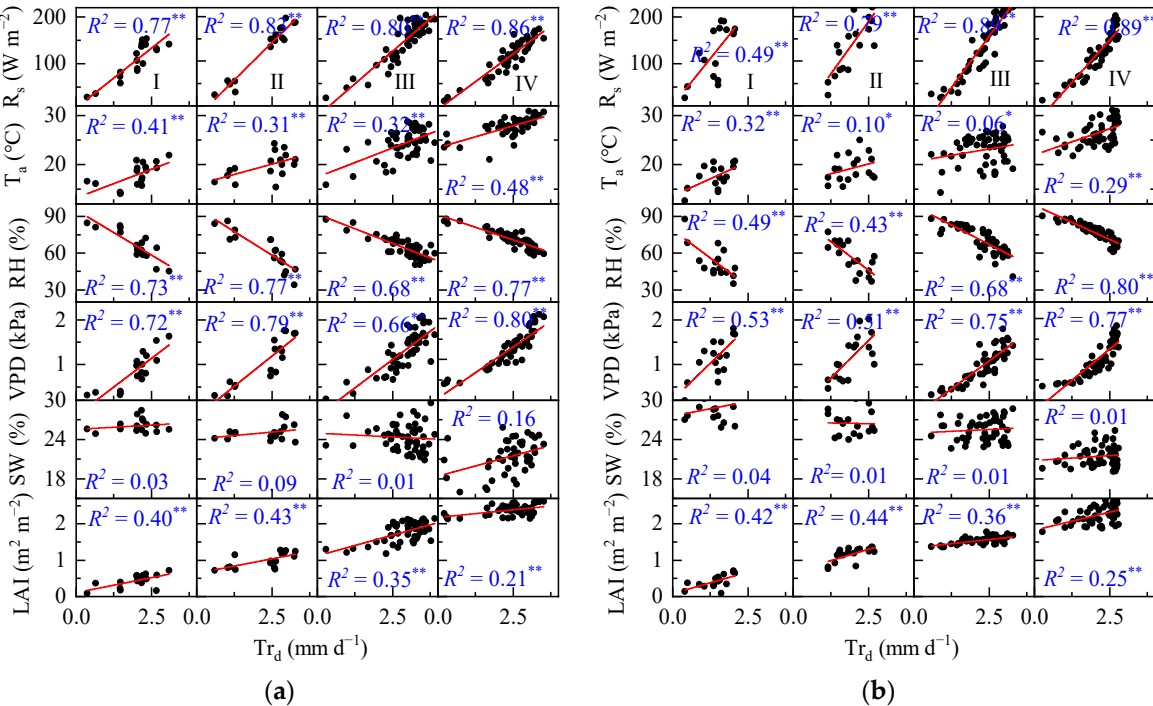

**Figure 4.** Correlation analysis between daily transpiration and environmental factors. (**a**) 2020; (**b**) 2021. * Indicates significant correlation between independent variable and dependent variable ($p < 0.05$); ** Indicates highly significant correlation between independent variables and dependent variables ($p < 0.01$). I, II, III, and IV respectively represent the new growth period, flowering and fruit setting period, fruit expansion period, and fruit maturation period.

### 3.3. Path Analysis of Instantaneous Transpiration Influence Factors in Grapes

As shown in Figures 3 and 4, the correlation between grape transpiration and meteorological factors was not consistent at different temporal scales. In addition, different environmental factors not only affect grape transpiration directly, but a single factor can also indirectly affect grape transpiration by affecting other factors. Therefore, it is necessary to systematically analyze the direct, indirect, and overall effects of environmental factors on grape transpiration at different temporal scales using path analysis.

Figure 5 shows the correlation between each meteorological factor on the daily scale, and the correlation between each meteorological factor and grape transpiration all reached the most significant level ($p < 0.01$). In terms of decision coefficients, in 2020, all four growth periods showed $R_s$ > VPD > $T_a$ > RH, and the decision coefficients ($R$) of $R_s$ were 0.88, 0.87, 0.90, and 0.90. In 2021, the new growth period showed $R_s$ > $T_a$ > VPD > RH. The flowering and fruit setting period and fruit expansion period showed similar results as those in 2020. The fruit maturation period showed $R_s$ > VPD > RH > $T_a$, with the $R$ of 0.69, 0.77, 0.88, and 0.89 for $R_s$ at the four growth periods. In addition, the $R$ also showed that $R_s$, VPD, and

$T_a$ were positively correlated with $F_s$, which promoted grape transpiration. While RH was negatively correlated with $F_s$, which inhibited grape transpiration. In terms of direct path coefficients (*P*), the influence factors of the new growth period and fruit expansion period in 2020 showed $R_s > T_a > RH > VPD$, and in the flowering and fruit setting period and fruit maturation period were $R_s > RH > T_a > VPD$ and $R_s > RH > T_a > VPD$. The *P* of $R_s$ with the largest direct effect in the four growth periods were 0.61, 0.69, 0.64, and 0.56, respectively. The results in 2021 were quite different from those in 2020. The new growth period was $R_s > VPD > RH > T_a$, the flowering and fruit setting period was $R_s > T_a > VPD > RH$, and the fruit expansion period and fruit maturation period were $R_s > RH > VPD > T_a$, where the *P* of $R_s$ in the four growth periods were 0.50, 0.80, 0.64, and 0.58, respectively. In terms of indirect path coefficients (*r*), VPD had the greatest indirect effect on $F_s$ through $R_s$, and the indirect path coefficients were 0.44, 0.49, 0.45, and 0.43, respectively. The results in 2021 were slightly different from those in 2020. Especially in the flowering and fruit setting period, $T_a$ had the largest indirect effect on $F_s$ through $R_s$ and *r* of 0.56. From the sum of indirect path coefficients, it can be seen that in 2020, the new growth period was $VPD > RH > T_a > R_s$, and the other three periods were $VPD > T_a > RH > R_s$. The sum of the indirect path coefficients of VPD with the largest indirect effect on $F_s$ through other factors were 0.80, 0.99, 0.81, and 1.17 in the four periods, respectively. In 2021, the new growth period showed $RH > T_a > VPD > R_s$, the flowering and fruit setting period showed $T_a > VPD > RH > R_s$, and the results of other growth periods were similar to those in 2020. Therefore, $R_s$ was the main control factor for the transient transpiration of grapes.

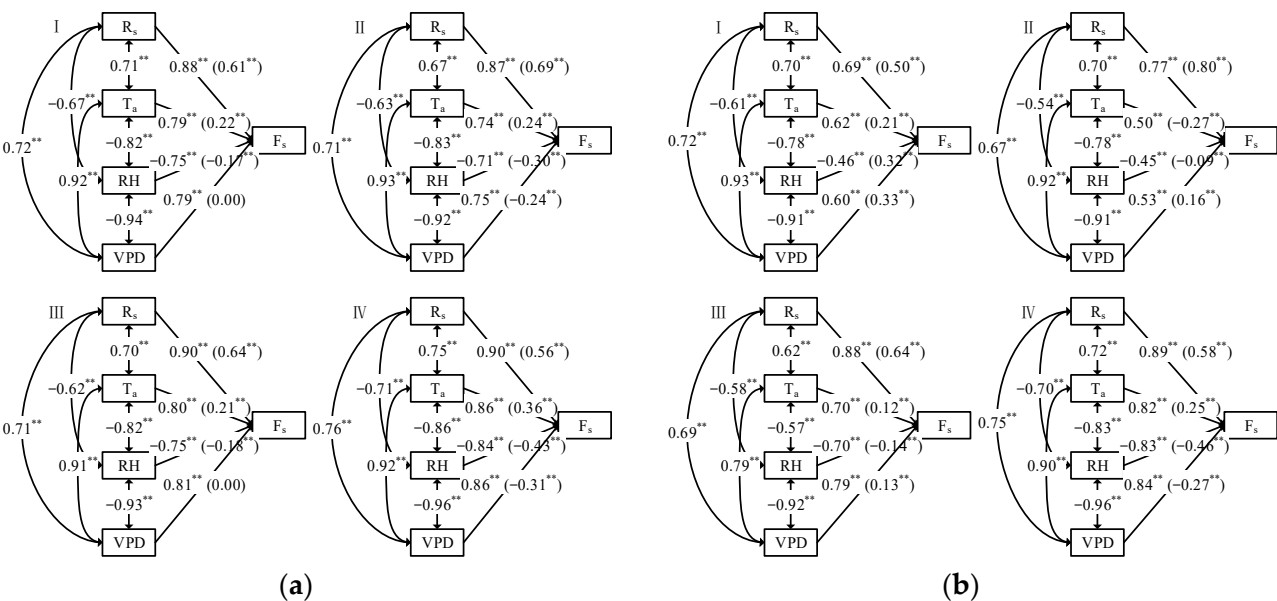

**Figure 5.** Path analysis results of instantaneous transpiration and meteorological factors in each growth period. (**a**) 2020; (**b**) 2021. I, II, III, and IV respectively represent the new growth period, flowering and fruit setting period, fruit expansion period, and fruit maturation period. * Indicates significant correlation between independent variable and dependent variable (*p* < 0.05); ** Indicates highly significant correlation between independent variables and dependent variables (*p* < 0.01).

### 3.4. Path Analysis of Daily Transpiration Influence Factors in Grapes

Figure 6 shows the correlation between each meteorological factor on the daily scale, and the correlation between each meteorological factor, LAI, and grape transpiration all reached the significant level (*p* < 0.05). However, the correlation between SW and various meteorological factors and grape transpiration reached the significant level only in certain growth periods, and there was no correlation with $Tr_d$. In terms of decision coefficients, the influencing factors in 2020 in the new growth period were $R_s > RH > VPD > T_a > LAI > SW$, $R_s > VPD > RH > LAI > T_a > SW$ in the flowering and fruit setting period, and $R_s > RH > VPD$

$> \text{LAI} > T_a > \text{SW}$ in the fruit expansion period. The results in the fruit maturation period were similar to those in the flowering and fruit setting period, where the *R* of $R_s$ were 0.88, 0.97, 0.90, and 0.93 in the four growth periods. The results in 2021 were slightly different from those in 2020, with $\text{VPD} > R_s > \text{RH} > \text{LAI} > T_a > \text{SW}$ at the new growth period, $R_s > \text{RH} > \text{VPD} > \text{LAI} > T_a > \text{SW}$ at the flowering and fruit setting period, $R_s > \text{VPD} > \text{RH} > \text{LAI} > T_a > \text{SW}$ at the fruit expansion period, and $R_s > \text{RH} > \text{VPD} > T_a > \text{LAI} > \text{SW}$ at the fruit maturation period. The *R* also showed that $R_s$, $T_a$, VPD, and LAI were positively correlated with $\text{Tr}_d$, which promoted grape transpiration, while the RH was negatively correlated with $\text{Tr}_d$, which inhibited grape transpiration. SW was negatively correlated with $\text{Tr}_d$ at some growth periods. In terms of direct path coefficients, each influence factor showed $\text{VPD} > T_a > R_s > \text{RH} > \text{LAI} > \text{SW}$ during the new growth periods in 2020, $R_s > \text{RH} > \text{VPD} > \text{SW} > T_a > \text{LAI}$ during the flowering and fruit setting period, $\text{Rs} > \text{RH} > \text{LAI} > \text{VPD} > \text{Ta} > \text{SW}$ during fruit expansion periods, and $\text{VPD} > \text{RH} > R_s > T_a > \text{LAI} > \text{SW}$ during the fruit maturation period. The factors with the greatest direct effect on $\text{Tr}_d$ at each growth period were VPD, $R_s$, $R_s$, and VPD, with *P* of 1.12, 0.78, 0.61, and −0.89, respectively. The results in 2021 were quite different from those in 2020, showing $\text{VPD} > T_a > \text{LAI} > \text{RH} > R_s > \text{SW}$ at the new growth periods, $\text{VPD} > T_a > \text{LAI} > \text{RH} > R_s > \text{SW}$ at the flowering and fruit setting period, $R_s > \text{RH} > \text{LAI} > \text{VPD} > T_a > \text{SW}$ at the fruit expansion periods, and the results at the fruit maturation period were similar to those in 2020. In terms of the indirect path coefficients, RH had the greatest indirect effect on $\text{Tr}_d$ through VPD in the new growth period in 2020. The indirect effect of VPD on $\text{Tr}_d$ through $R_s$ was the greatest at the flowering and fruit setting period. The indirect effect of RH on $\text{Tr}_d$ through $R_s$ was the greatest during fruit expansion periods. In the fruit maturation period, the indirect effect of RH on $\text{Tr}_d$ through VPD was the largest, and the *r* were −1.08, 0.67, −0.46, and 0.84 in the four growth periods, respectively. The results in 2021 differed from 2020: the new growth period was similar to that in 2020, the indirect effect of RH on $\text{Tr}_d$ through VPD was greatest at the flowering and fruit setting period, the indirect effect of VPD on $\text{Tr}_d$ through $R_s$ was greatest at the fruit expansion period, and the indirect effect of RH on $\text{Tr}_d$ through VPD was greatest at fruit maturation period. The *r* were −0.89, −0.58, 0.58, and 1.04, respectively. The sum of indirect path coefficients showed that the factors were $T_a > \text{RH} > R_s > \text{LAI} > \text{VPD} > \text{SW}$ during the new growth period in 2020, $\text{VPD} > \text{LAI} > T_a > \text{RH} > \text{SW} > R_s$ during the flowering and fruit setting period, $\text{VPD} > \text{RH} > T_a > \text{LAI} > R_s > \text{SW}$ during the fruit expansion period, and $\text{VPD} > \text{LAI} > \text{SW} > R_s > T_a > \text{RH}$ during the fruit maturation period. During the four growth periods, the factors with the largest indirect total effects were $T_a$, VPD, VPD, and VPD, and the sums of the indirect path coefficients were 1.17, 1.10, 0.98, and 1.78, respectively. The results of 2021 differed significantly from those of 2020, with $T_a > R_s > \text{SW} > \text{RH} > \text{LAI} > \text{VPD}$ during the new growth period, $T_a > R_s > \text{RH} > \text{LAI} > \text{VPD} > \text{SW}$ during the flowering and fruit setting period, $\text{VPD} > \text{RH} > \text{LAI} > T_a > R_s > \text{SW}$ during fruit expansion period, and $\text{VPD} > \text{LAI} > R_s > T_a > \text{SW} > \text{RH}$ during the fruit maturation period. Therefore, $R_s$ was the main control factor for the daily transpiration of grapes.

### 3.5. Path Analysis of Transpiration Influence Factors during the Whole Grape Growth Period

Table 2 shows the results of the path analysis of transpiration, with each factor at different temporal scales throughout the whole growth period. The effects of $R_s$, $T_a$, RH, and VPD on $F_s$ were all highly significant at the instantaneous scale ($p < 0.01$). In terms of decision coefficients, the factors were $R_s > \text{VPD} > T_a > \text{RH}$, with the *R* of $R_s$ reaching 0.86; in terms of direct path coefficients, the factors were $R_s > \text{VPD} > T_a > \text{RH}$, with the *P* of $R_s$ reaching 0.61. The total indirect effect of each factor is $\text{RH} > T_a > \text{VPD} > R_s$, and the indirect effect of VPD on $F_s$ through $R_s$ is the largest, with *r* of 0.43. On the daily scale, the effects of $R_s$, $T_a$, RH, VPD, and LAI on $\text{Tr}_d$ all reached a highly significant level ($p < 0.01$) while the effect of SW is not significant. In terms of decision coefficients, the factors were $R_s > \text{VPD} > \text{RH} > T_a > \text{LAI} > \text{SW}$, with the *R* of $R_s$ reaching 0.81. In terms of direct path coefficients, the factors were $\text{VPD} > R_s > \text{LAI} > \text{RH} > T_a > \text{SW}$, with the *P* of $R_s$ reaching 0.51. The total indirect effect of each factor was $\text{RH} > T_a > R_s > \text{VPD} > \text{SW} > \text{LAI}$. Among them, $R_s$ had the largest indirect effect on $\text{Tr}_d$ through VPD with an indirect path coefficient of 0.47. At the

growth period scale, the significance of $R_s$, $T_a$, RH, and VPD decreased, except for $T_a$ and LAI, which reached a significant level ($p < 0.05$) while the other factors were not significant. In terms of decision coefficients, the factors showed that $R_s$ > LAI > VPD > $T_a$ > SW > RH, among which the $R$ of $R_s$ reached 0.68. In terms of direct path coefficients, the factors showed that $R_s$ > LAI > RH > VPD > $T_a$ > SW, among which the $P$ of $R_s$ reached 0.61. The total indirect effect of each factor was $T_a$ > RH > VPD > $R_s$ > SW > LAI, with $T_a$ having the largest indirect effect on $Tr_p$ through LAI with an indirect path coefficient of 0.35. Therefore, $R_s$ was the main control factor of grape transpiration at all temporal scales throughout the whole growth period.

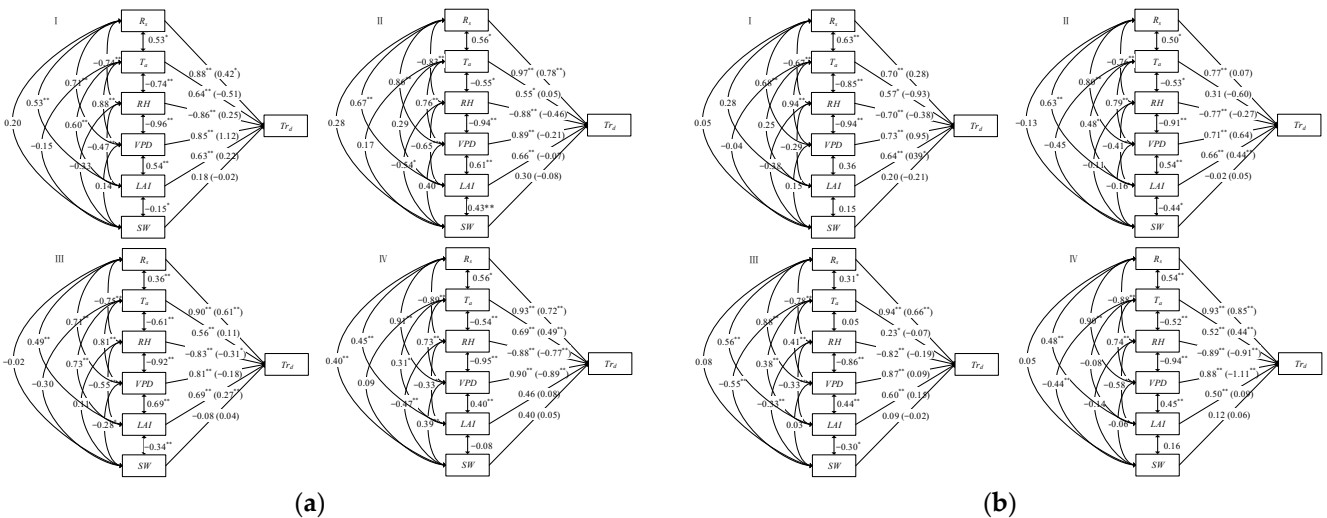

**Figure 6.** Path analysis results of instantaneous transpiration and meteorological factors in each growth period. (**a**) 2020; (**b**) 2021. I, II, III, and IV respectively represent the new growth period, flowering and fruit setting period, fruit expansion period, and fruit maturation period. * Indicates significant correlation between independent variable and dependent variable ($p < 0.05$); ** Indicates highly significant correlation between independent variables and dependent variables ($p < 0.01$).

**Table 2.** Path analysis results of transpiration and factors at different time scales in the whole growth period.

| Dependent Variable | Variable | Decision Coefficient | Direct Path | Indirect Path | | | | | | Total Indirect Path |
|---|---|---|---|---|---|---|---|---|---|---|
| | | | | $R_s$ | $T_a$ | RH | VPD | SW | LAI | |
| $F_s$ | $R_s$ | 0.86 ** | 0.61 ** | – | 0.04 | 0.00 | 0.21 | – | – | 0.25 |
| | $T_a$ | 0.65 ** | 0.07 | 0.35 | – | 0.00 | 0.23 | – | – | 0.58 |
| | RH | −0.65 ** | 0.00 | −0.35 | −0.03 | – | −0.26 | – | – | −0.65 |
| | VPD | 0.78 ** | 0.31 ** | 0.43 | 0.05 | 0.00 | – | – | – | 0.48 |
| $Tr_d$ | $R_s$ | 0.81 ** | 0.51 ** | – | −0.04 | −0.18 | 0.47 | 0.03 | 0.02 | 0.30 |
| | $T_a$ | 0.44 ** | −0.18 * | 0.12 | – | 0.03 | 0.31 | 0.24 | −0.07 | 0.62 |
| | RH | −0.53 ** | 0.26 * | −0.35 | −0.02 | – | −0.46 | 0.09 | −0.05 | −0.79 |
| | VPD | 0.78 ** | 0.61 ** | 0.39 | −0.09 | −0.20 | – | 0.08 | 0.00 | 0.18 |
| | LAI | 0.36 ** | 0.30 ** | 0.05 | −0.15 | 0.08 | 0.15 | – | −0.08 | 0.06 |
| | SW | −0.04 | 0.11 * | 0.08 | 0.12 | −0.12 | −0.01 | −0.21 | – | −0.15 |
| $Tr_P$ | $R_s$ | 0.68 | 0.61 | – | −0.09 | −0.12 | 0.16 | 0.05 | 0.07 | 0.07 |
| | $T_a$ | 0.34 * | −0.32 | 0.16 | – | 0.25 | 0.14 | 0.35 | −0.24 | 0.66 |
| | RH | −0.13 | 0.36 | −0.2 | −0.22 | – | −0.15 | 0.22 | −0.15 | −0.49 |
| | VPD | 0.52 | 0.35 | 0.27 | −0.12 | −0.15 | – | 0.22 | −0.05 | 0.17 |
| | LAI | 0.53 * | 0.51 | 0.05 | −0.22 | 0.16 | 0.15 | – | −0.13 | 0.02 |
| | SW | 0.19 | 0.26 | 0.16 | 0.29 | −0.21 | −0.06 | −0.26 | – | −0.07 |

Note: * Indicates significant correlation between independent variable and dependent variable ($p < 0.05$); ** Indicates highly significant correlation between independent variables and dependent variables ($p < 0.01$).

*3.6. Time Scale Effects of Transpiration Master Control Factors at Different Scales*

Figure 7 illustrates the variation of $R^2$ between grape transpiration and the main influencing factors at three temporal scales: instantaneous, daily, and growth period. Among them, the $R^2$ of RH and $T_a$ decreased obviously, and the $R^2$ at the growth period scale were only 0.02 and 0.12, respectively. The correlation between $R_s$ and grape transpiration

decreased with increasing scale but not significantly. $R_s$ is always the most important factor affecting grape transpiration in terms of instantaneous, daily, and growth period scales. In addition, this study also found that with the improvement of the research scale, the influence of LAI on transpiration was gradually enhanced, and its $R^2$ at the growth period scale reached 0.28, which exceeded other influencing factors except $R_s$.

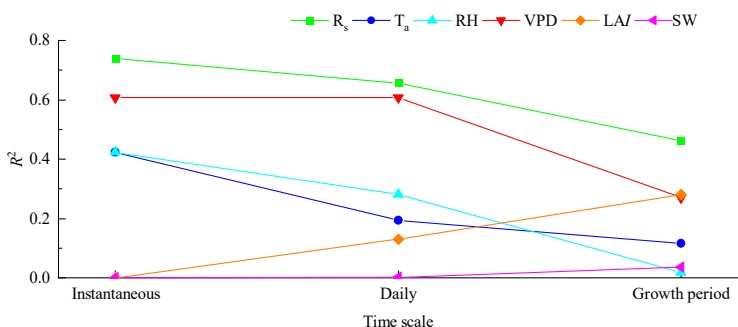

**Figure 7.** The determinants of each factor change with the increase of scale.

## 4. Discussion

### 4.1. Analysis of Main Control Factors of Instantaneous Transpiration

In this study, the correlation and path analysis of instantaneous transpiration and meteorological factors at different growth periods revealed that at the instantaneous scale, $R_s$, $T_a$, and VPD were significantly positively correlated with $F_s$ ($p < 0.05$). RH was significantly negatively correlated with $F_s$ ($p < 0.05$). Among them, the correlation between $R_s$ and $F_s$ is the best and the decision coefficient is the largest, which is the main control factor of instantaneous transpiration. However, the order of other influencing factors except $R_s$ varied in different years at different growth periods. This study also found that the direct effect of $R_s$ was the largest, but the order of the direct effect of other influencing factors varied in different years at different growth periods. The indirect effect on instantaneous transpiration and the indirect total effect on instantaneous transpiration also differed in different growth periods in different years. This is similar to the results of Cheng et al. [33], who concluded that the dominant meteorological factor affecting instantaneous transpiration is not unique, but the contribution of solar radiation is large. The main reason for the differences in results at different growth periods or in different years is that grape instantaneous transpiration is the result of a combination of internal and external factors. Internal factors include canopy structure, stomatal opening, trunk hydraulic structure, and root hydraulic transmission characteristics. External factors refer to environmental factors, mainly irrigation methods, soil moisture conditions, and meteorological factors [34]. In this study, only meteorological factors of transpiration were analyzed at the instantaneous scale, and the effects of internal factors such as canopy structure and stomatal opening on transpiration were not considered in different years and different growth periods. Therefore, the results were different for the results of different growth periods in different years. At some growth periods, due to high solar radiation, transpiration of the crop is intense, and the water uptake rate of the root system cannot be fully adapted to the water demand of transpiration, resulting in a short "siesta" phenomenon. This phenomenon was not considered in this study, which is also the reason for the differences in results at different growth periods [10]. In addition, continuous observation is difficult because LAI and SW vary less on the temporal scale. Therefore, the effects of LAI and SW at instantaneous scales were not considered in this study, which may also be the reason for the differences in results at different growth periods. Zhang et al. [35] found that at the instantaneous scale, the total effect of ground temperature on $F_s$ was the largest, and the direct effect of VPD on $F_s$ was the largest, which was different from this study. This may be due to species differences, or it may be due to differences in the growing seasons of winter wheat and grapes.

*4.2. Analysis of Main Control Factors of Daily Transpiration*

In this study, we found that the daily transpiration of grapes showed a single-peak trend during the whole growth period and reached the maximum value at the end of the fruit expansion period and the beginning of the fruit maturation period. This is different from the research of Li et al. [7] and Du et al. [34]. Du et al. found that the peak of daily transpiration occurred at the end of flowering and the beginning of fruit expansion, while Li et al. found that the maximum daily transpiration occurred at the mature period. This is mainly because the different site conditions, growing environment, water, and horticultural management measures of grapes can significantly affect grape transpiration.

In this study, the correlation and path analysis of daily transpiration and meteorological factors at different growth periods revealed that at the instantaneous scale, $R_s$, $T_a$, VPD, and LAI were significantly positively correlated with $Tr_d$ ($p < 0.05$), and RH was significantly negatively correlated with $Tr_d$ ($p < 0.05$). SW did not reach a significant level with $Tr_d$, and SW showed a negative correlation with $Tr_d$ at some growth periods. SW was negatively correlated with $Tr_d$ at some growth periods. Cai et al. [20] found that SW was positively correlated with $Tr_d$, which may be due to the fact that this experiment was fully irrigated and did not produce water stress. Zhang et al. [35] concluded that in the absence of soil water stress, the effect of soil water on evapotranspiration can be ignored. This study also found that, on the daily scale, $R^2$ and the decision coefficients were greater than $R_s$ for VPD during the new growth period in 2021, except for $R_s$, which showed the best correlation with $Tr_d$ and the largest decision coefficient and was the main control factor for daily transpiration. This is similar to the results of Gong et al. [10] and Li et al. [7]. However, Cai et al. [20] considered LAI as the main control factor of daily transpiration, followed by $R_s$, which is inconsistent with the results of this study. The main reason for this difference may be due to the different daily management measures of the crop. In this study, the changes in grape canopy structure and LAI were artificially intervened through pruning measures. It may also be due to the significant effect of insulation and warming in the greenhouse under heliostat growing conditions. The temperature at night is not only higher than the outdoor temperature but also increases rapidly and cools slowly, so the temperature is no longer a major factor restricting the liquid flow. At the same time, under greenhouse growing conditions, air convection was poor, and there is no obvious wind speed. This leads to the above conclusions differing from this paper. This study also found that the factors with the greatest direct, indirect, and total indirect effects on $Tr_d$ differed in different years at different growth periods. This is mainly because soil temperature is also an important factor in controlling the plant growth process and interacts with solar radiation to affect the water and heat conditions of the near-surface atmosphere, thereby affecting crop transpiration [35]. The effect of ground temperature on grape transpiration was not considered in this experiment and differed significantly between growth periods.

*4.3. Analysis of the Main Control Factors of Transpiration at the Growth Scale*

In this study, path analysis of transpiration and meteorological factors at the growth period scale revealed that the significance of $R_s$, $T_a$, RH, and VPD decreased at the growth period scale. Except for $T_a$ and LAI, which reached a significant level ($p < 0.05$), other factors were not significant. The total and direct effects of $R_s$ on transpiration were the largest, with decision coefficient and direct path coefficients reaching 0.79 and 0.61, respectively. The indirect effect of LAI on $Tr_p$ through $T_a$ was the largest, and RH had the largest indirect total effect. Thus, at the growth period scale, $R_s$ was the main control factor of grape transpiration. This is similar to the results of Nalevanková et al. [11], but the results of Zhang et al. [35] showed that soil temperature was the main controlling factor driving transpiration at the growth period scale. While the effect of soil temperature on transpiration was not considered in this study, some studies have shown that the effect of soil temperature on transpiration is not negligible [36,37]. Therefore, the effects of soil temperature on transpiration and the interaction between soil temperature and other meteorological factors need to be considered in subsequent studies to further reveal the time-

scale effect of transpiration. This study also found that the effects of meteorological factors such as Rs, Ta, VPD, and RH on transpiration gradually weakened, and the correlations of LAI gradually increased as the scale increased, and similar results were obtained by Wei et al. [9]. This is mainly because, in the growth of grapes, LAI first affects the land surface coverage and then affects the canopy transpiration surface area. The size and distribution of leaf area directly affect the interception and utilization of light energy by vegetation and thus affect the transpiration of vegetation.

## 5. Conclusions

Transpiration under greenhouse conditions is a complex process that is driven both by the characteristics of the plant and the greenhouse micro-environment. In this study, based on the measured data in 2020 and 2021, the transpiration and the influencing factors were analyzed at different time scales. The main findings of the study include: At the instantaneous scale, the effects of all meteorological factors on grape transpiration reached significant ($p < 0.05$) or highly significant ($p < 0.01$) levels at all growth periods. $R^2$ was between 0.25 and 0.84. At the daily scale, $R_s$, $T_a$, RH, VPD, and LAI were significantly correlated with transpiration ($p < 0.05$) at all growth periods, while SW had no correlation with transpiration. Except for the new growth period in 2021, the correlation between VPD and transpiration was the largest ($R^2 = 0.53$); other growth periods showed the best correlation between $R_s$ and transpiration, with $R^2$ ranging from 0.77 to 0.89. The results of the path analysis by growth period showed that the direct and total effects of $R_s$ on $F_s$ were the largest at all growth periods at the instantaneous scale, with decision coefficients ranging from 0.69 to 0.90. On the daily scale, the sum of the total effect of $R_s$ on daily transpiration was the largest, except for the new growth period, and the decision coefficients ranged from 0.70 to 0.94. The results of the whole growth period path analysis showed that $R_s$ had the greatest effect on instantaneous transpiration with decision coefficients of 0.86. At the daily scale, $R_s$ remained the most influential factor in grape transpiration with a decision coefficient of 0.81. On the scale of growth period, only $T_a$ and LAI were significantly correlated with transpiration ($p < 0.05$). The total effects of $R_s$ on transpiration were the largest, and the decision coefficient was 0.68. $R_s$ is always the most important factor affecting grape transpiration in terms of instantaneous, daily, and growth period scales. However, with an increase in time scale, LAI and SW have enhanced their effects on transpiration, while $R_s$, $T_a$, RH, and VPD have gradually weakened their effects on transpiration.

**Author Contributions:** Conceptualization, X.W. and D.P.; methodology, Y.B.; software, Y.L.; validation, D.P., X.W. and Y.L.; formal analysis, D.P.; investigation, S.J.; resources, X.W.; data curation, S.J.; writing—original draft preparation, D.P.; writing—review and editing, C.W. and X.W.; visualization, C.W.; supervision, X.W.; project administration, Y.B. All authors have read and agreed to the published version of the manuscript.

**Funding:** The study was funded by the Natural Science Foundation of Liaoning (No. 2021-MS-233), Liaoning provincial key research and development plan (NO. 2021JH2/10200022).

**Institutional Review Board Statement:** Not applicable.

**Informed Consent Statement:** Not applicable.

**Data Availability Statement:** Not applicable.

**Acknowledgments:** We acknowledge the staff of Shenyang Agricultural University for their technical support.

**Conflicts of Interest:** The authors declare no conflict of interest.

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
