# Peer review of "Path Analysis of the Main Control Factors of Transpiration in Greenhouse, Drip-Irrigated Grapes in Cold Areas of Northeast China"

_water, doi:10.3390/w14223764_

Round 1

Reviewer 1 Report

this manuscript tried to reveal the control factor of transpiration by path analysis, however when explain the result, some error occurs, and the authors are not so good at english writing. the experiment of 2 years is fine, here are some comments:

1.Line 178, we didnt see a unimodal trend for Trd and Rs

2. Figure 5a showed that Direct diameter coefficient ofâ… and â…¢ in 2020 is 0, then in line 260, it should be RH>VPD

3. In figure 5 and 6, I didnt find the indirect coefficient mentioned in 3.3 part.

4.line 290 , wrong order, should it be LAI>Ta>SW

5. The indirect coefficients in line 314 and 319 are not consistent with those given in Table 2.

6. line 424 should it be in daily scale?

7 Line 434: VPD has the largest decision coefficient, as shown in Figure 8.This experiment lacks many human factors. Although the author also mentioned, will these factors have an impact on the results?

9.In the last sentence of section 4.3, it seems that the discussion is not over. Why is this result?

10.Decision coefficients can be added when introducing the coefficients related to path analysis, and many decision coefficients are used in the result part.

11. abstract is too long , shorten it

12 in the conclusion part, conclude your results in one single paragraph instead of 5.

13 line 369-378, delete them, we have know your purpose in the above part , no need to repeat.

Reviewer 2 Report

Line 32: clarify what it means LAI.

Line 35: clarify what it means SW.

Line 112: 2.1. Study site, place a location map.

Line 126: indicate the characteristics of the dripper.

Line 127: What were they based on to establish those upper and lower humidity values?

Line 143: At what depth was the moisture measured?

Line 160:  Sewall Wright (1921), not in References.

Line 237: In addition, soil moisture has little variation in daily periods?

Line 490: to write "with" instead of "wirh".

Line 532: Referenes

consult also research carried out in other countries.

Round 2

Reviewer 1 Report

dear authors,

I think the manuscript is ok now